# Daptomycin Use for Persistent Coagulase-Negative Staphylococcal Bacteremia in a Neonatal Intensive Care Unit

**DOI:** 10.3390/antibiotics13030254

**Published:** 2024-03-12

**Authors:** Eleni Papachatzi, Despoina Gkentzi, Sotiris Tzifas, Theodore Dassios, Gabriel Dimitriou

**Affiliations:** 1Neonatal Intensive Care Unit, University General Hospital of Patras, 26504 Patras, Greece; elepapach@upatras.gr (E.P.); sotiristzifas@aol.com (S.T.); tdassios@upatras.gr (T.D.); gdim@upatras.gr (G.D.); 2Department of Pediatrics, University General Hospital of Patras, 26504 Patras, Greece

**Keywords:** late-onset sepsis, persistent bacteremia, CoNS, daptomycin

## Abstract

During the last two decades, the incidence of late-onset sepsis (LOS) has increased due to improved survival of premature neonates. Persistent bacteremia (PB) in LOS is defined as more than two positive blood cultures obtained on different calendar days during the same infectious episode. Although rare, PB should be treated aggressively to prevent adverse outcomes. Daptomycin, a lipopeptide antibiotic, has been used in neonates with persistent coagulase-negative staphylococci (CoNS) bacteremia with promising results, but studies reporting on the efficacy and safety of the agent are scarce. The purpose of this study was to evaluate the efficacy and safety of daptomycin use for persistent CoNS bacteremia in a neonatal cohort. This is a retrospective, observational, single-center study of neonates treated with daptomycin during 2011–2022 in the Tertiary Neonatal Intensive Care Unit (NICU) of the University General Hospital of Patras, Greece. For the years 2011–2022, there were 3.413 admissions to the NICU. During the last 3 years (2020–2022)—the active epidemiological surveillance period—123 infants (out of 851 admissions, 14.4%) developed CoNS bacteremia (LOS). During the study period, twelve infants with PB were treated with daptomycin. They had a median gestational age of 32 weeks (IQR 31–34) and mean (SD) birth weight of 1.840 (867) grams. CoNS bacteremia isolates were *s. epidermidis* (50%), *s. haemolyticus* (20%), *s. hominis* (20%) and *s. warneri* (10%). The decision to start daptomycin (6 mg/kg/dose twice daily) was taken on median day 10 (ΙQR 7–15) of infection. None of the infants had focal complications or meningitis. Daptomycin therapy caused no renal, hepatic, muscular or gastrointestinal adverse events. One neonate developed seizures, and one death occurred due to multiple complications of prematurity. Most infants (11/12) were successfully treated and eventually had negative blood culture. Daptomycin monotherapy showed an adequate cure rate in premature neonates with persistent CoNS bacteremia in a tertiary NICU. In our study, daptomycin was effective and well tolerated; the safety profile, however, needs to be confirmed in larger studies and randomized controlled trials.

## 1. Introduction

Late-onset sepsis (LOS) is defined as an episode of sepsis after the first 72 h of life and is related to increased morbidity and mortality in the Neonatal Intensive Care Unit (NICU) [1]. Incidence of LOS varies geographically between 0.6 and 14% [2]. During the last two decades, the incidence of LOS has increased in many countries, and in particular, the incidence of coagulase-negative staphylococci (CoNS) LOS has almost doubled, with a reported mortality around 8% in all gestational ages [3,4]. Of note, 45% of neonates who eventually died due to LOS had bacteria resistant to the empiric therapy given [5]. Risk factors for LOS include administration of antenatal steroids, prematurity and longer hospital stay, prolonged mechanical ventilation and central line/peripheral line catheterization [4]. According to several studies, CoNS sepsis may be associated with adverse neurodevelopmental outcomes and prolonged hospital stay [2,6,7].

During the last decade, CoNS have emerged as the predominant pathogens of LOS in NICUs, with prevalence between 53.2 and 77.9% [7,8]. In the past, most septicemias caused by CoNS were thought to be endogenous; however, many strains are considered as opportunistic pathogens in immunocompromised patients [9]. *s. epidermidis* (the commonest CoNS isolate), and more precisely specific genotypes (SSCmec, the ica gene cluster and the insertion sequence element IS256) [10], are highly adaptable to the NICU environment and can develop resistance to antiseptics used and produce biofilm. Biofilms are responsible for difficult-to-treat infections and persistent bacteremias—especially in neonates [11]. Risk factors for sepsis due to CoNS include extreme prematurity, very low birth weight, indwelling catheters, prolonged total parenteral nutrition and others [12].

Persistent bacteremia is defined as more than two positive blood cultures obtained on different calendar days during the same infectious episode, caused by the same isolate on blood cultures [8]. Persistent bacteremia due to CoNS has been linked to biofilm formation and subsequent antibiotic resistance [9,10]. Treatment of persistent bacteremia due to CoNS should be prompt and aggressive, and escalation of treatment should be assessed as early as possible to prevent adverse outcomes [11].

Treatment options for persistent bacteremia due to CoNS include vancomycin monotherapy or vancomycin plus rifampicin. Rifampicin adds intracellular bactericidal activity and achieves higher concentration in biofilms due to its lipophilic characteristics [12,13,14]. During recent years, biofilm-producing CoNS are developing an increasing resistance to vancomycin. More precisely, while using vancomycin, many researchers have observed an increase in the optimal trough vancomycin level (from 10–20 to 15–20 mg/L) required to clear bacteremias, and in some cases of neonatal LOS, vancomycin fails to clear CoNS bacteremias (despite optimal trough vancomycin levels) [15]. Decreased vancomycin susceptibility with increased minimum inhibitory concentration (MIC) has been reported for CoNS, and this could be attributed to pressure from previous antibiotic exposure. For these reasons, there is a clinical need for further treatment options, alternative to vancomycin.

Linezolid can be used as a second-line treatment for persistent CoNS bacteremias. Linezolid has excellent CSF penetration and ubiquitous tissue distribution [16]. However, in neonates, it should be used with caution, under specific circumstances, due to its bacteriostatic effect and the risk of myelosuppression in prolonged courses [17].

Daptomycin is a cyclic lipopeptide antibiotic with dose-dependent bactericidal activity. It works by depolarization of the bacterial cell membrane, inhibiting the synthesis of DNA, RNA and proteins [14]. It is licensed in adults and children (above one year of age) with bloodstream infections caused by *staphylococcus aureus* (methicillin-susceptible and methicillin-resistant isolates) and in adults with right-sided infective endocarditis [18]. Additionally, daptomycin’s licensed use includes skin infections (for adults and children one to seventeen years of age) and complicated skin infections caused by *s.aureus* (including methicillin-resistant isolates), *streptococcus pyogenes*, *streptococcus agalactiae*, *streptococcus dysgalactiae* subspecies *equisimilis* and *enterococcus faecalis* (vancomycin-susceptible isolates only). Off-label uses of daptomycin include cerebrospinal fluid shunt infection, diabetic foot infection, endocarditis, intracranial or spinal epidural abscess, bacterial meningitis, osteomyelitis, discitis, prosthetic joint infection and septic arthritis [15]. As per the antibiotic’s Summary of Product Characteristics (SPC), its use in neonates is off-label due to possible musculoskeletal, neuromuscular and nervous system adverse effects that have been observed in canine models. In the literature, there are very limited data available for its safety profile in neonates. Since daptomycin’s action is dose-dependent, the specific neonatal dose that minimizes the risk of adverse side effects has yet to be confirmed. Recent studies have shown that a neonatal dose of 6 mg/kg twice daily (versus 10 mg/kg once daily—adult dose) is efficient to treat associated infections (across different causative organisms) [16,17]; however, there are no pharmacokinetic data available in the neonatal population. It is possible that higher doses would be needed in neonates compared to adults and children because of the relatively larger neonatal volume of distribution and faster renal clearance of the drug.

The purpose of this study was to evaluate the efficacy and safety of daptomycin use for persistent CoNS bacteremia in a neonatal cohort.

## 2. Results

### 2.1. General

During the study period (2011–2022), there were 3.413 admissions to the Neonatal Intensive Care Unit. During the years 2020–2022 (active epidemiological surveillance), there were 123 neonates (out of 851 admissions) with culture-proven LOS (123/851, 14.4%) due to CoNS. Total isolates were 131 in 123 patients (some neonates had more than one episode of LOS). Of the total 131 CoNS isolates, 47 caused persistent bacteremia (47/131, 35.8%). During the study period, twelve neonates required daptomycin to clear infection caused by CoNS. Distribution of patients in the years of the study was as follows: 2011 (1), 2012 (2), 2013 (3), 2016 (1), 2019 (1), 2020 (2) and 2022 (2). Patient demographics are presented in Table 1. Causative pathogens are presented in Figure 1, and patient characteristics per CoNS isolate in Table 2.

During the years 2020–2022, the incidence of CoNS isolates causing persistent bacteremia increased from 31.5% (2020) to 35.9% (2021) and 42% (2022). The incidence of culture-proven neonatal sepsis per total number of admissions was 18.3% in 2020, 13.6% in 2021 and 11.9% in 2022. The incidence of early-onset sepsis (cases per admission) was 2.6%, 0.7% and 0.3% in 2020, 2021 and 2022, respectively. The incidence of LOS was 15.7%, 12.9% and 11.6% in 2020, 2021 and 2022, respectively. More precisely, LOS incidence per isolate in blood culture (not per patient) was 24%, 14.3% and 14.5% in 2020, 2021 and 2022, respectively.

### 2.2. Daptomycin-Treated Neonates

For the neonates treated with daptomycin, the antimicrobials administered, duration of treatment per course (days) and causative pathogen (per patient) are presented in Figure 2. The decision to treat with daptomycin was taken on median day 10 of infection (ΙQR 7–15). The dose of daptomycin was 6 mg/kg intravenously every twelve hours. Most patients managed to clear CoNS bacteremia while on daptomycin (for more than 24 h). While receiving daptomycin, the most difficult pathogen to clear (among CoNS) was *s. epidermidis.* Interestingly, there was one neonate that had three positive blood cultures (*s. epidermidis*) while on daptomycin (Table 2).

On the first day of LOS, all neonates had a peripheral line, whereas two had also a central line. The majority (11/12) were enterally fed, six were receiving invasive ventilation, and four noninvasive ventilation. There were no focal complications (abscess formation, infectious endocarditis, etc.). None of the neonates had positive cerebrospinal fluid culture or urine culture, and none had meningitis. There were no renal or hepatic impairments while or after taking daptomycin (up to discharge). There was no eosinophilia nor eosinophilic pneumonia in our cohort. One neonate developed seizures without a history of intraventricular hemorrhage. This was a 32-weeks-of-gestation male neonate that developed LOS due to *s. haemolyticus* at 9 days of life, had previously completed a course of ampicillin and gentamicin for EOS, and for CoNS LOS, he received linezolid for 4 days, which was then changed to daptomycin for 14 days. On day 10 of daptomycin use, he developed clinical seizures that required phenobarbital. Later, his MRI revealed a congenital cause (small focal cortical dysplasia).

Furthermore, in our cohort, the majority (11/12) of infants were discharged, and one passed away due to multiple complications of prematurity. This was an ex-25 weeker (birth weight 770 g) who suffered from persistent pulmonary hypertension and had multiple complications during his stay in the NICU (pneumothorax; pulmonary hemorrhage; infective fungal endocarditis; bronchopulmonary dysplasia, which required prolonged ventilation and finally tracheostomy; persistent ductus arteriosus; and periventricular leukomalacia). This neonate died at a corrected gestational age of 5.5 months (267 days old) and was never discharged from the NICU.

Neurodevelopmental follow-up showed normal neurodevelopmental outcomes in eight neonates (8/11), mild neurodevelopmental impairment in one neonate (1/11), and two neonates (2/11) did not attend planned follow-up.

## 3. Discussion

This study highlights that the use of daptomycin was associated with high efficacy in treating late-onset infections by coagulase-negative staphylococci in the neonatal intensive care environment and that the use of this agent was not associated with significant adverse events or side effects.

During the years of active epidemiological surveillance (2020–2022), the rates of persistent bacteremia remain similar (5–6% cases per admission), but more and more LOS episodes tended to be persistent bacteremias (28.6% in 2020 versus 34.1% in 2021 and 43% in 2022). There are limited published data regarding the incidence of persistent neonatal bacteremia, but the rates observed in our cohort are alarming compared to our previous study (2016–2017, 25%)—although this study referred to cases and not isolates [19]—or to other centers (8.5%) [20]. Because of this trend observed, and in order to decrease persistent bacteremia incidence, infection control measures were taken (central line indications were revised, checklist for central line insertion was updated, and hand hygiene compliance was improved through weekly training and monitoring). Our neonatal team continuously monitor the incidence of persistent bacteremia due to CoNS and possible associations. Persistent bacteremia was caused by *s. epidermidis*, followed by *s. haemolyticus*, *s. hominis* and *s. warneri*. Τhis trend agrees with already published studies and could be attributed to neonatal flora of *s. epidermidis* in mucocutaneous sites and the nasopharynx during the first week of life [21]. Mean gestational age and birth weight were not different between the above categories, but infants within the *s. epidermidis* category had the highest number of positive blood cultures.

Daptomycin can cause skeletal muscle myopathy (elevated creatine kinase, CPK), which is reversible upon its discontinuation [22,23]. However, in our study, we did not observe this phenomenon. More precisely, daptomycin was effective and well tolerated in our population. There were no adverse effects observed such as renal, hepatic or gastrointestinal impairment. Creatinine phosphokinase (CPK) was normal during and after daptomycin treatment. There are limited data in the literature for the use of daptomycin and its safety profile in neonates (the majority are case reports), and the only pharmacokinetic data available refer to adults or children above one year of age.

In a study from Ohio, USA, researchers reported the successful use of daptomycin in an ex-24 weeker who suffered from methicillin-resistant *staphylococcus epidermidis* bacteremia and impaired renal function. The dose used was 6 mg/kg per dose intravenously every 12 h, and the course duration was 12 days. There were no adverse effects observed during therapy or after completion [24]. In another study from Leicester, United Kingdom, the authors described a case of a twin born at 27 weeks of gestation who eventually developed methicillin-resistant *s. aureus* bacteremia and was successfully treated with daptomycin. The dose used was 10 mg/kg once daily, and it was given for 14 days. There were no adverse events observed [25]. Moreover, a case report from Greece presented a case of an ex-27 weeker with a bloodstream infection caused by *s. epidermidis* that was successfully treated with daptomycin (including drug level monitoring), with no subsequent complications related to daptomycin. The dose used was 6 mg/kg, and the duration of the treatment was 17 days [26].

Daptomycin in neonates has also been reported to be safe even in more prolonged courses. In a study from Padua, Italy, the authors presented a case report of an ex-24 weeker with infective endocarditis (blood and central catheter tip cultures showed positive results for both methicillin-resistant *s. epidermidis* and *s. capitis*), which was successfully treated with daptomycin at a dose of 6 mg/kg every 12 h for 6 weeks. There were no renal, hepatic or neurodevelopmental adverse events observed until follow-up at 5 months of age [27]. There were similar findings from a case report describing an ex-28 weeker with infective endocarditis due to methicillin-resistant *staphylococcus aureus*, which was successfully treated with daptomycin. The dose used was 6 mg/kg, and the course lasted for 40 days [16].

Recently, in a case series study from Riyadh, Saudi Arabia, out of 21 neonates that received daptomycin (the dose was either 6 mg/kg every 12 h or 10 mg/kg every 24 h), eight neonates died. Mean alanine aminotransferase (ALT) was higher after starting daptomycin, and there were no muscular or neurological complications observed. The causes of death were abdominal bowel perforation (one neonate), Gram-negative sepsis (three neonates) or Gram-positive sepsis (four neonates). Of those who died, four neonates suffered from infective endocarditis (three Gram-positive and one Gram-negative), one had bacteremia due to a Gram-negative organism, and one had early-onset sepsis caused by a Gram-positive organism. In the infective endocarditis group, the higher mortality rates while taking daptomycin were attributed to the rapid progression of sepsis to intractable congestive heart failure which was established prior to the initiation of daptomycin [28].

In a report of an ex-25 weeker with *s. haemolyticus* bacteremia, daptomycin use at a dose of 6 mg/kg every 12 h caused severe hepatotoxicity and DIC, and the neonate died 48 h post-daptomycin initiation [29].

In our cohort, the decision to start daptomycin was taken on median day 10 of infection. The dose of daptomycin was 6 mg/kg intravenously every twelve hours. The daptomycin course was around 10 days in most patients, and the most difficult to treat cases were those caused by *s. epidermidis*. This has been observed in the past by other authors and could be related in many cases to biofilm formation [19,30,31,32]. In a previous study from our department, most of the *s. epidermidis* isolates causing persistent bacteremia in neonates were producing biofilm (compared to non-persisting isolates), and this was the most significant determinant for the development of persistent bacteremia [19].

Daptomycin is a licensed drug for *s. aureus* bacteremia in children, and its safety profile has been adequately monitored in many studies and randomized controlled trials [33,34,35,36,37]. Our study complements the literature by studying a group of more premature infants—compared to the existing published data—with persistent bacteremia. Our data indicate a safe profile of daptomycin for use in neonates with LOS and persistent bacteremia due to CoNS. Our population did not have any focal complications or meningitis. Daptomycin, at our reported regimen, was found to be adequate to clear bacteremia and was well tolerated. The course of daptomycin used as monotherapy was 10 days from the first negative blood culture.

At this point, the authors would like to highlight the phenomenon of heteroresistance that has been observed in some staphylococci isolates (*s. aureus*) in patients that have previously received vancomycin and are treated with daptomycin; there is the possibility of decreased susceptibility to daptomycin. However, at present, daptomycin resistance is relatively uncommon in pediatric populations, and to our best knowledge, it has not been reported in neonates [38,39].

## 4. Materials and Methods

### 4.1. Patients

This is a retrospective observational study including all neonates with persistent bacteremia caused by CoNS who received treatment with daptomycin from January 2011 up until December 2022 in the Neonatal Intensive Care Unit of the University General Hospital of Patras in Greece (tertiary-level unit). The study was approved by the Ethics Committee of the University General Hospital of Patras (806/11.12.2018). Informed parental consent was waived as the patient data were analyzed anonymously. All neonates had more than two positive blood cultures (caused by the same CoNS isolate) in the same episode of sepsis, after 72 h of age. The blood samples were derived by peripheral venipuncture using aseptic technique. Demographics recorded were sex, gestational age, prematurity, birth weight, mode of delivery, risk factors for sepsis (group B *streptococcus* colonization; premature rupture of membranes, PROM), Apgar scores, intubation at birth, hospital transfer, admission temperature, congenital anomalies, day of life at the episode of LOS and presence of peripheral or central line. Prematurity was defined as gestational age less than 37 weeks. PROM was defined as rupture of membranes more than 18 h prior to delivery. All neonates had culture-proven sepsis including clinical and laboratory criteria. Clinical criteria included temperature instability, respiratory deterioration (increase oxygen requirements, need for ventilatory support and/or increased number of apneas—desaturations), circulatory deterioration (hypotension and signs of peripheral hypoperfusion) and/or feeding intolerance (abdominal distension or significant aspiration of gastric residue that required a decrease of >20% or cessation of feeding for at least 24 h). Laboratory criteria included positive blood cultures (more than two) and any of the following: platelet count < 150,000/mm^3^, and/or changes in blood glucose > 50% and/or CRP > 1 mg/dL (normal range < 0.5 mg/dL). Sepsis work-up included full blood count, CRP, electrolytes, renal and liver function, blood culture, urine culture and cerebrospinal fluid culture.

In CoNS bacteremia, first-line treatment consisted of vancomycin (±rifampcin if still bacteriemic after 48 h). The vancomycin dose was adjusted to achieve a trough serum concentration within the therapeutic range (15–20 mcg/mL). When the sensitivities tests (antibiogram) were available from the microbiological department, antibiotic regimen was reevaluated according to the minimum inhibitory concentration (MIC). Blood culture turnaround time was 24–48 h. The decision to escalate treatment to third line (linezolid) and daptomycin (final regimen) was taken considering the e-tests (MIC), the presence of clinical or laboratory deterioration or a persistent bacteremia despite targeted treatment and acceptable trough levels of vancomycin.

All neonates were monitored for adverse effects while receiving antimicrobials. The full blood count was monitored every 48 h to measure the white blood cell count, neutrophils, platelets, hemoglobin and hematocrit. Renal (urea, creatinine) and liver function (alanine transaminase, aspartate transaminase, creatine kinase, lactate dehydrogenase, bilirubin) were also monitored. Physical examination and regular neurodevelopmental assessments were performed daily while inpatient and at follow-up (at 12 months of age).

### 4.2. Active Epidemiological Surveillance (Years 2020–2022)

Active epidemiological surveillance was the day-to-day surveillance/monitoring of sepsis incidence, presence of risk factors for LOS, antimicrobial use and NICU capacity, and was conducted by the authors as a baseline infection control activity in the NICU.

### 4.3. Microbiological Evaluation

Blood cultures were processed in the microbiology laboratory of our hospital, into BacT/ALERT PF pediatric FAN vials of the BacT/ALERT 3D system for aerobic and anaerobic bacteria (bioMerieux SA, Marcy l’ Etoile, France). Coagulase-negative staphylococci were identified based on colony morphology, Gram stain, and positive catalase and negative coagulase test results (Slidex Staph Plus; bioMerieux, Tokyo, Japan). Species identification was performed with the VITEK 2 System (bioMerieux). When phenotypic identification showed efficacy lower than 99%, a molecular method based on the tuf gene was applied. Tests for susceptibility to anti-staphylococcal agents were performed by the disk diffusion method and e-test (AB Biodisk, Solna, Sweden), according to CLSI guidelines.

### 4.4. Statistical Analysis

The variables were checked for normality of distribution using visual inspection of their distribution curves. Quantitative variables were described either by mean and standard deviations if normally distributed or by median and interquartile range if non-normally distributed. Categorical variables were described as absolute and relative frequencies. Due to small sample size, descriptive data analysis was performed using Microsoft^®^ Excel^®^ for Microsoft 365 MSO (Version 2401 Build 16.0.17231.20236) (Microsoft Corporation, Redmond, WA, USA).

## 5. Conclusions

In conclusion, daptomycin monotherapy in treating persistent bacteremia due to coagulase-negative staphylococci was found to be effective and well tolerated in neonates hospitalized in a tertiary Neonatal Intensive Care Unit. Clinicians may consider using this antibiotic when dealing with persistent bacteremias. More trials, ideally randomized controlled ones, are needed to further investigate the safety and efficacy profile as well as the pharmacokinetics of daptomycin in this vulnerable population.

## Figures and Tables

**Figure 1 antibiotics-13-00254-f001:**
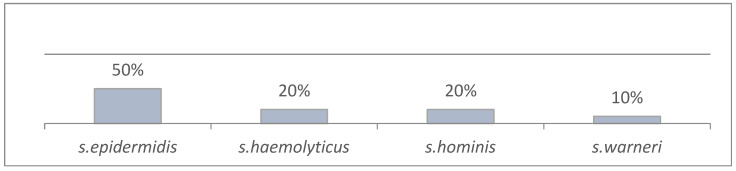
CoNS isolates in persistent bacteremia (2011–2022) (*n* = 47).

**Figure 2 antibiotics-13-00254-f002:**
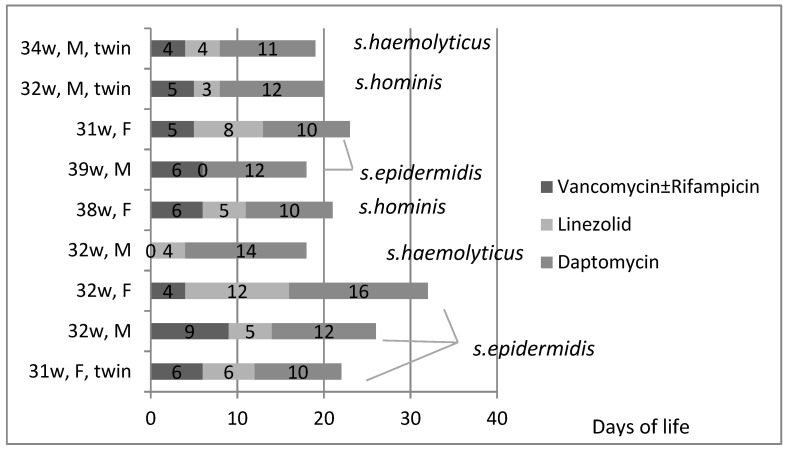
Antibiotic course, total days of treatment and causative pathogen (per patient per course) in neonates with persistent bacteremia due to CoNS, during 2011–2022. X axis corresponds to days of life.

**Table 1 antibiotics-13-00254-t001:** Patient demographics in neonates with persistent CoNS bacteremia for the years 2011–2022 in NICU, Patras, Greece.

Demographics (1)	
Median gestational age	32 weeks (IQR 31.25–33.5)
Mean birth weight	1840 ± 867.4 g
Apgar 1 min (median IQR)	9 (8–9)
Apgar 5 min (median IQR)	9 (9–10)
Mean day of life of LOS (±sd)	7 ± 3.9
Admission temperature (°C)	36.5 ± 0.65
Demographics (2)	Number/Total (%)
Sex (male)	8/12 (66.7%)
Prematurity (<37 weeks)	8/10 (80%)
Vaginal delivery	3/10 (30%)
Emergency caesarean section	7/10 (70%)
PROM > 18 h	2/10 (20%)
Meconium-stained liquor	1/10 (10%)
Intubation at birth	6/10 (60%)
Singleton pregnancy	8/12 (66.7%)
Congenital anomalies	1/12 (8.3%)
Ex utero transfers	2/12 (16.7%)

**Table 2 antibiotics-13-00254-t002:** Patient demographics in neonates with persistent CoNS bacteremia (*n* = 47), per CoNS isolate, for the years 2011–2022. (*) in the same patient.

Demographics (3)	*s. epidermidis*	*s. haemolyticus*	*s. hominis*
Median gestational age (IQR) weeks	33 (31–39)	33 (32–34)	34 (31–38)
Mean birth weight (±SD) grams	1930 ± 1127	1990 ± 325	1990 ± 551
Number of positive blood cultures	61	15	15
Positive blood cultures while taking daptomycin (>24 h)	3 *	0	0
Mean NICU admission (days)	46	30	27

## Data Availability

The original contributions presented in the study are included in the article, further inquiries can be directed to the corresponding author.

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
