# Peer review of "Daptomycin Use for Persistent Coagulase-Negative Staphylococcal Bacteremia in a Neonatal Intensive Care Unit"

_antibiotics, 2024, doi:10.3390/antibiotics13030254_

Round 1

Reviewer 1 Report

Comments and Suggestions for Authors

The study seems well conducted and is very useful as daptomycin is not yet licensed for use under one year of life. Collecting experiences on use in neonatal age is very important.

The purpose of the study should be written at the end of the introduction and also in the abstract

The species of the microorganism must always be written in lower case: check line 170 and check all the text.

Line 159 We have demonstrated that the use of daptomycin was associated with high efficacy ... The study did not prove... it only highlighted. I think it's better to remove "We have demonstrated that"

Author Response

Dear Reviewer,

We would like to thank you for your response and the suggested corrections for our manuscript. We have highlighted the suggested changes in the manuscript attached. More precisely:

  • the purpose of the study has been added in the introduction and in the abstract as suggested
  • species of the microorganisms are now written in lower case (correction in the whole manuscript and in the figures/graphs)
  • line 159 was rephrased and the phrase ''we have demonstrated'' was removed

Looking forward to your reply to the revised manuscript.

Yours sincerely,

Eleni Papachatzi

Reviewer 2 Report

Comments and Suggestions for Authors

 Introduction: please verify the off label use of daptomycin as described at line 86-87. Daptomycine is approved for the treatment of complicated skin infections, bacteremia and right sided edocartidis caused by S. Aureus references  N.Engl J Med. 2006 ; 355(7): 653-665- Am J Med 2007 ; 120 (supl 1) :  S28-S33

Results: specify in a better way what kind of active surveillance is : line102

Discussion: specify how the authors calculate  the incidence of LOS episodes line 164

described if there was case of eosinophilia e/o  eosinofilic pneumoniae as adversed events line 175

described with more details why  there was  an increase  incidence of CONs during the study period line 170-171-172

described if the authors takes some infection control measures against CONs incidence increase

Author Response

Dear Reviewer,

We would like to thank you for your response and the suggested corrections for our manuscript. We have highlighted the suggested changes in the manuscript attached. More precisely:

  • Daptomycin use is not approved for neonates and its use in our cohort is off label. In lines 86-87 we describe other off label uses of daptomycin that have been described in the literature.
  • ''Active surveillance''. Thank you very much for this comment, it was not explained in the text so we added further details in line 299-302 in Methods.
  • There was no eosinophilia nor eosinophilic pneumonia (added as requested, line 149)
  • It is not clear why there was observed an increase incidence for CONS persistent bacteremia during these years. In is highlighted (line 176) that there are very few studies in the literature describing persistent bacteremia incidence and prevalence in the neonatal setting. However, authors are currently monitoring the incidence of LOS (PB) due to CONS and possible risk factors (central line presence, peripheral line presence,  lines insertion technique,  antimicrobial use, NICU capacity etc) in order to identify possible correlations. The above was added in line 182.
  • Infection control activity taken was added in lines 178-181, again thank you for this comment.

We are looking forward to your response.

Your sincerely,

Eleni Papachatzi

Reviewer 3 Report

Comments and Suggestions for Authors

Dear Authors,

Your work is highly valued and very well presented.

I have two comments:

1. Provide subsections in Results - General (Line 101 to 130) and Daptomycin-treated neonates. Please provide additional information, on how are your 12 patients distributed over the research period of 2011-2022. In the Daptomycin subsection, please include a table with more information about the characteristics of your 12 patients, like time spent in NICU, antibiotics used, gestational age, birth weight, etc.

 2. Line 272, serum concentration should be mcg/mL. 

Author Response

Dear Reviewer,

We would like to thank you for your response and the suggested corrections for our manuscript. We have highlighted the suggested changes in the manuscript attached. More precisely:

  • We have added subsections in the results as suggested
  • We described how our patients were distributed in the study period (line 111-112)
  • In the daptomycin subsection there is a table with patient characteristics (gestational age and birthweight --> table 1, antibiotics used --> figure 2). We moved the table from the general part 2a to daptomycin subsection 2b. Regarding the time spent in NICU, these data are not presented as gestational ages differ and the data are not comparable. NICU stay per isolate is described in table 2.
  • Line 272 mcg/ml was corrected - it was a typo (l instead of μ)

We are looking forward to your response.

Yours sincerely,

Eleni Papachatzi